# First Isolation and Phylogenetic Analyses of Tick-Borne Encephalitis Virus in Lower Saxony, Germany

**DOI:** 10.3390/v11050462

**Published:** 2019-05-21

**Authors:** Mathias Boelke, Malena Bestehorn, Birgit Marchwald, Mareike Kubinski, Katrin Liebig, Julien Glanz, Claudia Schulz, Gerhard Dobler, Masyar Monazahian, Stefanie C. Becker

**Affiliations:** 1Institute for Parasitology, University of Veterinary Medicine Hannover, Bünteweg 17, 30559 Hanover, Germany; mathias.boelke@tiho-hannover.de (M.B.); mareikekub@gmail.com (M.K.); Katrin.Liebig@tiho-hannover.de (K.L.); Julien.Glanz@fli.de (J.G.); claudia.schulz@live.de (C.S.); 2Research Center for Emerging Infections and Zoonoses, University of Veterinary Medicine Hannover, Bünteweg 17, 30559 Hanover, Germany; 3Parasitology Unit, University of Hohenheim, Emil-Wolff-Straße 34, 70599 Stuttgart, Germany; Malena1Bestehorn@bundeswehr.org (M.B.); GerhardDobler@bundeswehr.org (G.D.); 4Institute of Microbiology of the Bundeswehr, Neuherbergstraße 11, 80937 Munich, Germany; 5The Governmental Institute of Public Health of Lower Saxony (NLGA), Roesebeckstraße 4-6, 30449 Hannover, Germany; parasitologie@nlga.Niedersachsen.de (B.M.); Masyar.Monazahian@nlga.Niedersachsen.de (M.M.)

**Keywords:** tick-borne encephalitis virus, tick, phylogenetic analysis, surveillance, field study

## Abstract

Tick-borne encephalitis (TBE) is the most important tick-borne arboviral disease in Europe. Presently, the main endemic regions in Germany are located in the southern half of the country. Although recently, sporadic human TBE cases were reported outside of these known endemic regions. The detection and characterization of invading TBE virus (TBEV) strains will considerably facilitate the surveillance and assessment of this important disease. In 2018, ticks were collected by flagging in several locations of the German federal state of Lower Saxony where TBEV-infections in humans (diagnosed clinical TBE disease or detection of TBEV antibodies) were reported previously. Ticks were pooled according to their developmental stage and tested for TBEV-RNA by RT-qPCR. Five of 730 (0.68%) pools from *Ixodes* spp. ticks collected in the areas of “Rauher Busch” and “Barsinghausen/Mooshuette” were found positive for TBEV-RNA. Phylogenetic analysis of the whole genomes and E gene sequences revealed a close relationship between the two TBEV isolates, which cluster with a TBEV strain from Poland isolated in 1971. This study provides first data on the phylogeny of TBEV in the German federal state of Lower Saxony, outside of the known TBE endemic areas of Germany. Our results support the hypothesis of an east-west invasion of TBEV strains in Western Europe.

## 1. Introduction

Tick-borne encephalitis (TBE) is one of the most important zoonotic central nervous system (CNS) diseases in humans in Europe and Asia with 10,000 to 15,000 cases per year [1]. TBE in humans can result in severe neurological symptoms such as meningitis (60%), meningoencephalitis (30%) and meningoencephalomyelitis (10%) with potential lethal (0.2%–1%) outcome of the disease [2,3]. The causative agent, TBE virus (TBEV) belongs to the genus *Flavivirus* in the Flaviviridae family. Humans generally become infected with TBEV following the bite of an infected tick. Mainly in eastern European countries, the alimentary transmission of the virus via contaminated dairy products is another possible cause of TBEV-infection. TBE viruses are divided into five subtypes, whose distribution is geographically linked to the occurrence of their main vector tick species: the European subtype (transmitted mainly by *Ixodes ricinus*) and the Siberian and Far-Eastern subtypes (both transmitted mainly by *I. persulcatus*) and two new subtypes, the Baikalian (isolated from *I. persulcatus*) and the Himalayan subtype (tick vector unknown) [4,5,6,7]. To date, only strains of the European subtype have been identified in Germany [8]. The vector tick for TBEV in Germany is the castor bean tick *I. ricinus*. The sylvatic cycle of TBEV, involves ticks and wild vertebrate hosts, in particular small rodents [1,9]. Ticks play the most important role for maintaining TBEV in nature acting both as vector and as reservoir. Due to the short viremia in humans and the absence of detectable virus in cerebrospinal fluid of patients, most TBEV isolates are obtained from questing ticks in so-called TBEV foci [9].

The TBEV genome consists of a positive single-stranded RNA molecule (+ssRNA) of approximately 11 kilobases in length containing one open reading frame (ORF) and encoding for a large polyprotein (about 3400 amino acids) [1]. This polyprotein is cleaved into three structural proteins (C = capsid, M = membrane and E = envelope) and seven non-structural proteins (NS1, NS2A, NS2B, NS3, NS4A, NS4B, NS5), which are necessary for virus replication [10]. The E protein acts as the major surface protein of the virus interacting with cell receptors of tick vectors and mammalian hosts and mediating fusion of the virus with the cell membrane. The virus neutralizing immune response in the mammalian host via antibodies is mediated against the E protein [11].

In Germany, most endemic areas are located in the southern federal states of Bavaria and Baden-Wuerttemberg [12]. Sporadic human TBEV cases have also been reported outside of these endemic areas, e.g., the federal state of Lower Saxony [13]. Since TBE was classified as a notifiable disease in Germany in 2001, 61 human cases have been reported (2001–2018) in the federal state of Lower Saxony [14] (see Appendix A). Detection and characterization of invading TBEV strains will add information to our knowledge on the distribution and spread of TBEV and herewith facilitate the surveillance and understanding of this important disease. Here, we describe the detection, isolation and first phylogenetic data of TBEV strains detected in ticks in the German federal state of Lower Saxony.

## 2. Materials and Methods

### 2.1. Virus Screening

In 2018, 4798 questing ticks (704 adults, 4094 nymphs) were collected by flagging the low vegetation at five different sampling sites located at two distinct areas in Lower Saxony, namely “Rauher Busch” (N52°34′; E8°48′) and “Barsinghausen/Mooshuette” (N52°19′; E9°23′). Both locations had been associated with humans that showed clinical TBE disease and/or were seropositive for TBEV. Ticks were pooled according to stage and sampling site and stored at 4 °C for a maximum of 7 days until further processing. Each of the analyzed pools contained 5 to 10 adults or 10 to 20 nymph ticks. Pools were homogenized using steel beads in 500 µL cell culture medium (Leibowitz’s L-15 or MEM Eagle, Thermo Scientific, Waltham, MA, USA) and TissueLyser II (Qiagen, Hilden, Germany). The homogenates were clarified by centrifugation and total RNA was extracted from the homogenates using the NucleoSpin^®^ RNA Virus kit (Macherey-Nagel, Dueren, Germany) or MagNA Pure96 Viral RNA Small Volume Kit (Roche, Mannheim, Germany) according to the manufacturer’s instructions. The extracted RNA was re-suspended in RNase-free water and screened for TBEV-RNA by quantitative reverse transcription-PCR (RT-qPCR) following the protocol developed by Schwaiger and Cassinotti [15]. TBEV-RNA of the Austrian Neudoerfl strain (U27495.1) was used as a positive control, while RNase-free water served as negative control. Furthermore, the quality of RNA extraction was tested in randomly selected samples (10% of all pools) using a RT-qPCR targeting the 16S ribosomal RNA (rRNA) of *I. ricinus* (Schwaiger and Cassinotti [12]).

### 2.2. Virus Cultivation

We inoculated A549 cells (ATCC^®^ CCL-185™) with 100 µL aliquots of TBEV-RNA positive tick homogenate (diluted 1:10 in minimal essential medium (MEM)). After 1 h of incubation at 37 °C, the virus homogenates were discarded, cells were washed three times with PBS and 4 mL of MEM supplemented with 2% fetal bovine serum (FBS) and antibiotics (Penicillin/Streptomycin Pan Biotech; Aidenbach, Germany, Gentamicin/Amphotericin Thermo Fisher, Waltham, MA, USA) were added. Negative controls were inoculated with MEM. This procedure was performed for all five TBEV-RNA positive pools. The homogenates TBEV-LS-Rauher Busch P16, -Rauher Busch P19 and TBEV-LS-Barsinghausen/Mooshuette P51 were done in Hannover at the laboratory of the Research Center for Emerging Infections and Zoonoses, the isolates TBEV-LS-Barsinghausen/Mooshuette-HB IF06 8040 and TBEV-LS-Barsinghausen/Mooshuette-HB IF06 8033 were generated at the laboratory of the Bundeswehr Institute of Microbiology in Munich.

### 2.3. PCR and Phylogenetic Analysis of TBEV Isolates

For the TBEV isolates TBEV-LS-Barsinghausen/Mooshuette-HB IF06 8040 and –HBIF06 8033, and TBEV-LS-Rauher Busch P19 the whole genome was generated. Therefore, two 5.6. kb PCR fragments, covering the whole genome, were generated using primers TBEw-1/TBEw-c5670/TBEw-5451/TBEw-c11141 of Andersen et al. [16] in a conventional RT-PCR (SuperScript™ III Reverse Transcriptase; Invitrogen, Karlsruhe, Germany). The PCR products were mechanically sheered using a Bioruptor UCD-200 (Diagenode Diagnostics, Liège, Belgium) and prepared for sequencing using the TruSeq Nano DNA Low Throughput Library Prep Kit (Illumina, Inc., San Diego, CA, USA). The Illumina MiSeq platform was used for sequencing and the sequencing chemistry used was the MiSeq reagent kit V3 (Illumina, Inc., San Diego, CA, USA), according to the manufacture’s protocol. De novo assembly of the sequences was performed using the software Spades v.3.12 [17]. The maximum likelihood tree (Figure 1) was generated using the general time reversible model with five categories. The G+ parameter used was 0.3942. The neighbor joining tree (not shown) was calculated using the initial parameters from MEGA6.0 (https://www.megasoftware.net). For statistical support, 1000 bootstraps were calculated for each tree.

A 1488 bp fragment of the E gene was amplified from the different TBEV isolates using the primer pairs TBE-885/TBE-c2571a/b of Kupča et al. [5] and a conventional RT-PCR kit (SuperScript™ III Reverse Transcriptase; Invitrogen, Karlsruhe, Germany) according to the manufacturer’s instructions. Sanger sequencing of the resulting PCR fragment was conducted using the TBE-885/TBE-c2571a/b primers as well as the TBE-c1648 primer [5] (GATC Biotech Ag, Konstanz, Germany). E gene sequences were obtained by *de novo* assembly of the chromatogram files using Geneious 9.1.5 (https://www.geneious.com). The sequences of the E genes for TBEV-LS-Rauher Busch-P16 (MK903683), -P19 (MK903681) and TBEV-LS-Barsinghausen/Mooshuette-P51 (MK903682), -HB IF06 8040 (MK903679) and –HB IF06 8033 (MK903680) as well as available sequences of the E gene from the NCBI GenBank database (http://www.ncbi.nlm.nih.gov/blast/Blast.cgi) and so far unpublished sequences of the German National Reference Laboratory (TBEV Jaegerhof JH98/13 (MK903684); TBEV-RG18-1971-POL (MK903685)) were used for phylogenetic analysis with the MEGA6.0 software (https://www.megasoftware.net).

## 3. Results and Discussion

Of the 730 tested pools, we found five pools (0.68%) positive for TBEV-RNA. At the collection site “Rauher Busch”, two pools with adult ticks and at the site of “Barsinghausen/Mooshuette” one and two pools with adults and nymphs, respectively, were positive for TBEV-RNA. The amplification of 16S rRNA was successful in all tested pools indicating efficient RNA extraction and minimizing the probability of false negative results due to inhibition of the PCR reaction by tick products. The minimum infection rates (MIR) for TBEV in ticks from Lower Saxony were 1.05% in adults and 0.45% in nymphs, which is similar to the MIR for TBEV described in Sweden in 2008 (0.55% in adults and 0.23% in nymphs) [18], Poland and Lithuania in 2006–2009 (0.24% in adults and 0.11% in nymphs) [19] and in the district of Passau in Bavaria, Germany, in 2001 (0.03%–6.38% in adults and 0.08%–1.1% in nymphs) [20] and 1997–1998 (0.9%–2.0% for all tick stages) [21]. The MIR of nymph and adult tick stages from endemic foci in the Black Forest region near Freiburg in Baden-Wuerttemberg, Germany, were slightly higher in 1997 (2.9%–3.4%) and in 1998 (0.6%–1.1%) [21]. Similarly, a MIR of 4.48% in adults and 0.51% in nymphs was reported at the well described TBEV focus Torö island, south-east of Stockholm, in 2008 [18]. These data suggest that TBEV prevalence considerably varies by year and location, most probably due to small sample sizes analyzed and due to the frequency of sampled individual tick stages [22]. However, our MIR in Lower Saxony lies well within the published range.

To confirm the presence of infective virus from the tick samples, all homogenates were inoculated on A549 cells. The three homogenates TBEV-LS-Rauher Busch P16 and 19, and TBEV-LS-Barsinghausen/Mooshuette P51 done at the Research Center for Emerging Infections and Zoonoses caused a cytopathic effect (CPE) beginning 3–4 days post infection (p.i.). RT-qPCR analyses of the cell culture supernatants collected at 6–7 days p.i. revealed positive results for TBEV-RNA in the samples TBEV-LS-Rauher Busch-P16, TBEV-LS-Rauher Busch-P19 and TBEV-LS-Barsinghausen/Mooshuette-P51. Hereinafter, these three virus isolates are referred to as TBEV-LS-Rauher Busch-P16, TBEV-LS-Rauher Busch-P19 and TBEV-LS-Barsinghausen/Mooshuette-P51. Two additional viruses that were isolated at the Bundeswehr Institute of Microbiology in Munich are referred to as TBEV-LS-Barsinghausen/Mooshuette-HB IF06 8040 and TBEV-LS-Barsinghausen/Mooshuette-HB IF06 8033.

The genome sequences obtained during this study (TBEV-LS-Barsinghausen/Mooshuette isolates HB IF06 8033 (MK922615) and 8040 (MK922616) and TBEV-LS-Rauher Busch P19 (MK922617)) were used for the construction of a phylogenetic tree using the complete open reading frame of these and all available sequences from NCBI GenBank (Figure 1).

Based on the full genome sequences, the new isolates TBEV-LS-Rauher Busch P19 and TBEV-LS-Barsinghausen/Mooshuette are closely related to each other and more distantly related to other German isolates where full genome sequences are available such as Salem, K23, MucAr and AS33. The closest relationship is seen to a cluster including Russian, Korean, Finnish, South Bohemian and the Salem sequences. In addition, we compared the sequences of the open reading frame of the two isolates TBEV-LS- Rauher Busch P19 and TBEV-LS-Barsinghausen/Mooshuette HBIF06 8033 to each other (Appendix A). This comparison revealed 36 nucleotide difference between the two sequences resulting in 10 amino acid exchanges. This again points towards a close genetic relationship of these isolates as compared to the study by Kupča et al. [5] which describes the relationship of the isolate AS33 and Salem showing 251 nucleotide differences resulting in 26 amino acid exchanges between those two strains.

For a further in-depth characterization of the virus isolates TBEV-LS-Rauher Busch-P16 and -P19 as well as TBEV-LS-Barsinghausen/Mooshuette-P51 HB IF06 8040 and HB IF06 8033, we used a PCR assay described by Kupča et al. [5] to amplify the viral E gene. The E gene is the most commonly used marker for phylogenetic classification of TBEV subtypes and strains and most sequence data are available for this gene sequence [23]. The maximum likelihood and neighbor joining trees generated by phylogenetic analysis for the E gene (Figure 2) compared with the respective trees of the whole coding sequences (Figure 1) resulted in a similar dendrogram. The close genetic relationship of the E gene sequences from TBEV isolates in Lower Saxony Barsinghausen/Mooshuette and Rauher Busch despite geographic distance between those two sampling locations indicate that the two isolates have a common ancestor and have spread locally. This is unusual as two natural TBE foci separated by a distance of about 60 km usually have genetically more differentiated viruses [24]. The close genetic relationship implies that the TBEV strains were recently distributed from one location to the other, and so far had no time to adapt and evolve in a focus-specific TBE genetic clade.

Furthermore, the E gene sequences of TBEV-LS-Rauher Busch-P16 and -P19 as well as TBEV-LS-Barsinghausen/Mooshuette-P51, -HB IF06 8040 and –HB IF06 8033 clustered with TBEV strains from Battaune in the German federal state of Saxony (TBEV-S-Battaune-DZIF16 1198; DZIF16 1205, DZIF17 20 and DZIF18 119) [25] and Lodz in Poland (kindly provided by Pawel Steffanoff) (Figure 2). The Polish isolate was detected in 1971 and might therefore be an ancestor for the Battaune and the Lower Saxony isolates. The proposed east-west spread is in accordance with one of the three main hypotheses for TBEV spread. According to molecular phylogenetic data of E gene sequences, TBEV appeared around 2500 years ago in the Far East and then spread to West Eurasia [23,24,26]. This hypothesis is however contrasted by other studies. For example, analyses of whole genome sequences suggest Western Siberia as an origin for TBEV and a subsequent bi-directional spread of TBEV to West and East Eurasia [27]. Nevertheless, on the smaller regional scale, the general east-west concept is supported by all published data collected so far [24,26,28].

Regarding the means of TBEV spread, there are several possibilities: (i) continuous spread: terrestrial transport of TBEV-infected ticks by infestation on wild animal hosts such as roe deer or wild boar [29,30], (ii) discontinuous spread: aerial transport of TBEV-infected ticks by avian hosts, mostly associated with stepping stones of migratory bird species [31], or anthropogenic transport including transport of infected animals [22]. For the recent TBEV isolates from the upper Rhine valley detected in 2016–2018, terrestrial and aerial spreading pattern by wild animals seem to apply [32]. Weidmann et al. [24] discussed discontinues spread of TBEV via anthropogenic routes to explain the phylogenetic relationship between TBEV isolates from the Czech Republic (South Moravia) and Bavaria (Haselmühl) in Germany. The two Czech and German regions where the respective TBEV isolates were detected are separated by a mountain range of 1400 m altitude; hence, making TBEV-spread by terrestrial hosts rather unlikely. Similarly, the close phylogenetic relationship between the TBEV-LS-Barsinghausen/Mooshuette/Rauher Busch and TBEV-S-Battaune strains with the TBEV-Lodz-RG18-1971-Pol strain from Lodz in Poland (a distance of 750 km via the E30 and A2 motorways or of 870 km via the E40 and A4 motorways in Poland and Germany, respectively; see Appendix A) might also reflect a direct transportation of TBEV-infected ticks between both locations by motorized vehicles. The E30 and E40 motorways in Poland are highly trafficked by trucks for freight transports between Germany and Poland. In 2014, these two motorways were the second most important routes for goods transferred to Germany with an annual volume of 25 million tons [33]. Interestingly, the second route via the E40 passes by Battaune in Saxony - thus, supporting the hypothesis of a common introduction route for TBEV-LS-Barsinghausen/Mooshuette/Rauher Busch and TBEV-S-Battaune strains (Appendix A). In a Russian study it has been hypothesized that the Baltic TBEV strains of the Siberian and Far Eastern subtypes were exported from their Russian areas of distribution in Siberia along the routes of the trans Baikalian highway and the trans Baikalian railway [34]. However, the currently available data do not allow final conclusions on the origin of the TBEV strains from Lower Saxony. More isolates from other locations along the E30 or E40 would be needed to either confirm or disprove the hypothesis of freight transport as an anthropogenic route for TBEV distribution.

Taken together, the detection of five different TBEV strains from two different locations in Lower Saxony highlights the increased prevalence of this virus in German regions that have been formally classified as non-endemic for TBEV. Accordingly, the increased TBEV prevalence demands an increase of public awareness raising and surveillance efforts for TBEV in Lower Saxony and possibly the other northern federal states of Germany. Furthermore, the first phylogenetic characterization of TBEV strains from Lower Saxony suggests a classical east-west invasion event as indicated by the high similarity of virus strains from Saxony and Lower Saxony, supporting the existing hypothesis of TBEV distribution patterns.

## Figures and Tables

**Figure 1 viruses-11-00462-f001:**
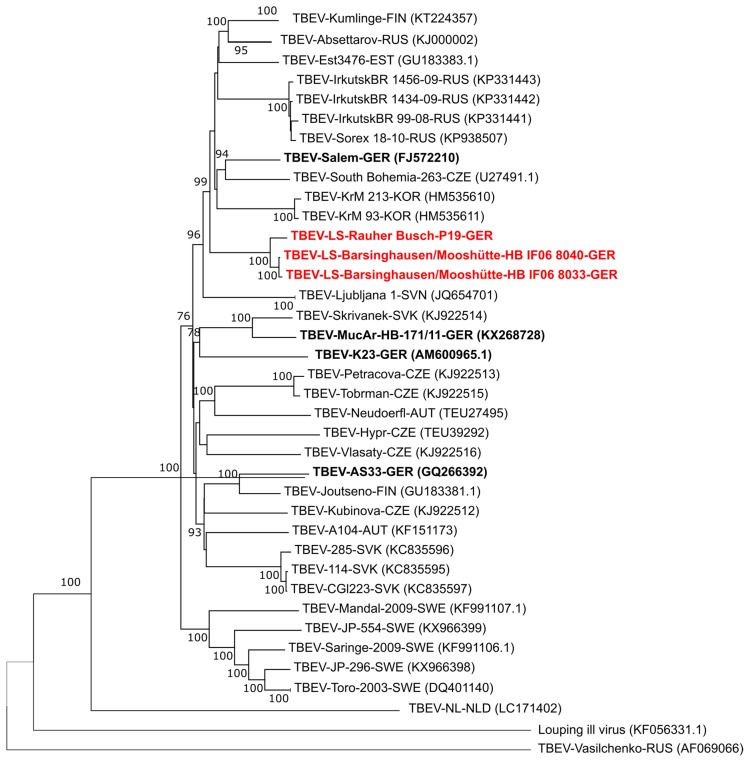
Phylogenetic tree of whole genome sequences available in GenBank and the two TBEV-LS-Barsinghausen/Mooshuette isolates HB IF06 8033 (MK922615) and 8040 (MK922616) and TBEV-LS-Rauher Busch P19 (MK922617) obtained in the present study (marked red). Virus isolate names are given as follows: TBEV-isolate name or number-three letter country code according to UN standard (accession number for the sequences as available from NCBI). UN country codes are: AUT: Austria; CHE: Switzerland; CZE: Czech Republic; EST: Estonia; FIN: Finland; GER: Germany; KOR: Korea; NLD: Netherlands; RUS: Russian Federation; SVN: Slovenia; SVK: Slovakia; SWE: Sweden. All sequences obtained from German isolates are marked bold, the new sequences obtained during this study are additionally marked in red. As an outgroup the Louping ill virus (KF056331.1) and TBEV-Vasilchenko (AF069066) were used.

**Figure 2 viruses-11-00462-f002:**
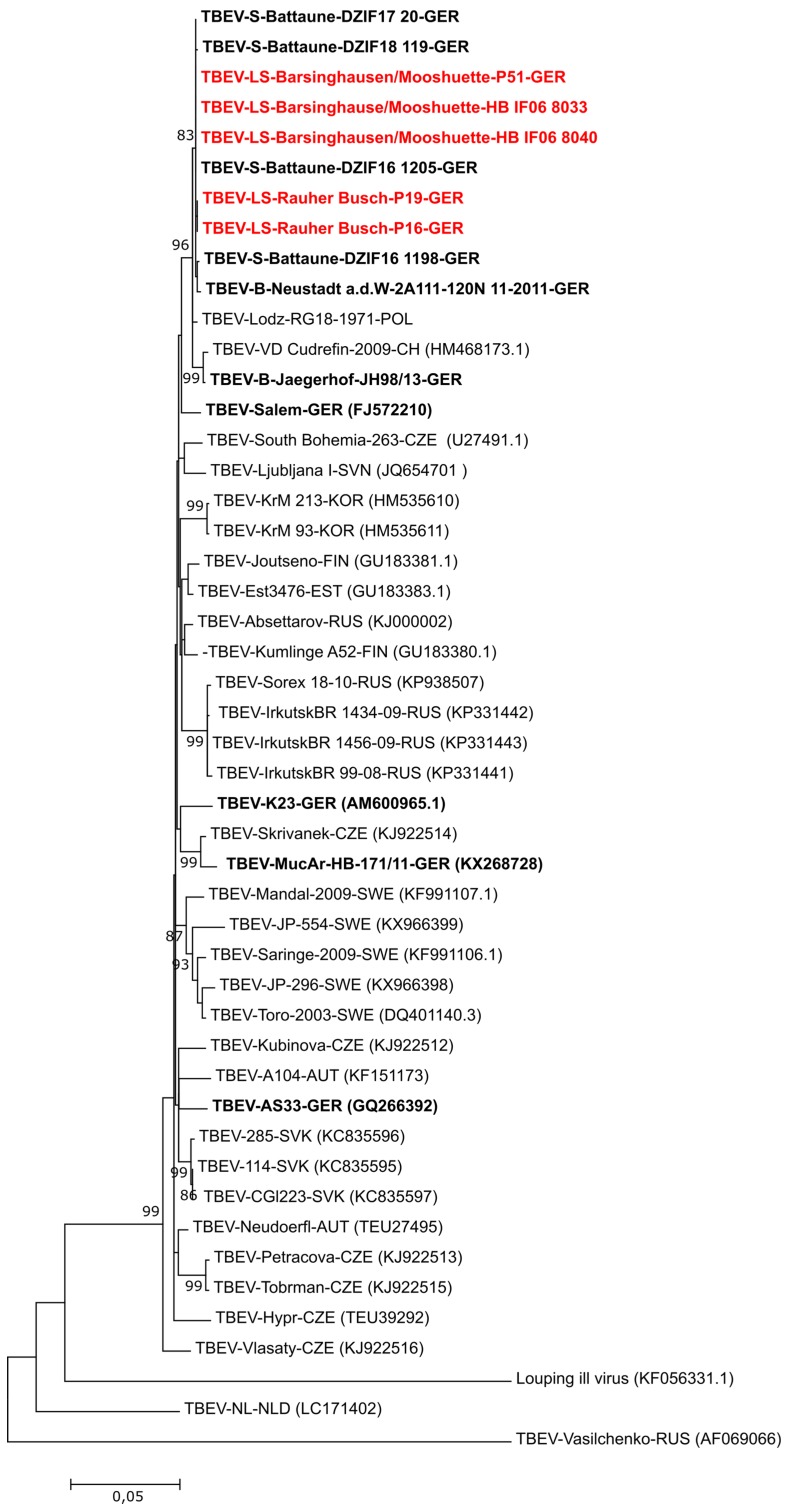
Phylogenetic tree (Maximum likelihood Tree, MEGA6.0) of the five “TBEV-LS” isolates collected in “Barsinghausen/Mooshuette” and “Rauher Busch” (TBEV-LS-Rauher Busch-P16 (MK903683), -P19 (MK903681) and TBEV-LS-Barsinghausen/Mooshuette-P51 (MK903682), -HB IF06 8040 (MK903679) and –HB IF06 8033 (MK903680) in the German federal state of Lower Saxony, 2018, using E gene sequences. Virus isolate names are given as follows: TBEV-isolate name or number-three letter country code according to UN standard (accession number of the sequence as available from NCBI). UN country codes are: AUT: Austria; CHE: Switzerland; CZE: Czech Republic; EST: Estonia; FIN: Finland; GER: Germany; KOR: Korea; NLD: Netherlands; POL: Poland; RUS: Russian Federation; SVN: Slovenia; SVK: Slovakia; SWE: Sweden. As an outgroup strain Louping ill virus (KF056331.1), TBEV-NL (LC171402) and TBEV-Vasilchenko (AF069066) were used. All sequences obtained from German isolates are marked bold, the new sequences obtained during this study are additionally marked in red.

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
