# Peer review of "First Isolation and Phylogenetic Analyses of Tick-Borne Encephalitis Virus in Lower Saxony, Germany"

_viruses, 2019, doi:10.3390/v11050462_

Reviewer 1 Report

Boelke et al. report the spread of TBEV into a new region. Similar spreading events are closely followed and reported elsewhere in Europe, too. This type of surveillance allows updated risk assessments to be made concerning this important arboviral disease. Research in this field also provides data on which further predictions of disease spread can be done.

Overall, I find this study well executed, however, I think much can be done to clarify the results and their significance. I will also recommend further sequence analysis.

Line 52: Please check the sentence starting “Ticks paly the…” and rephrase.

Lines 65-67: How many cases are reported in the other states? I would suggest adding a map illustrating the variation in incidence/annual case numbers. I went on to check the database, but I found it quite tricky to find the actual data, but I do think this piece of information is essential for understanding the significance of these results.

Methods/TBEV-isolates: It is a bit difficult to follow the isolation process, and what was exactly done on which tick pool, so please check the parts 2.2 and 2.3 concerning this. There are five pools positive for TBEV-RNA. For two isolates, you generated the whole genome. Why only two and why these two? Are these the two isolated elsewhere and what was the protocol used there? The three isolates described in this study were successful and from an isolate, the whole genome can and should be obtained (unless you can specifically state why not). Please also add a short description of the new sequences, how similar are they to each other and to other TBEV-Eur strains.

Figure 1. Please show clearly the newly obtained sequences and other sequences from Germany as well.

Figure 2. This is a highly selected set of sequences, which may have an impact on the result. Please recalculate this phylogenetic tree with a larger collection of available E-gene sequences.

Author Response

Boelke et al. report the spread of TBEV into a new region. Similar spreading events are closely followed and reported elsewhere in Europe, too. This type of surveillance allows updated risk assessments to be made concerning this important arboviral disease. Research in this field also provides data on which further predictions of disease spread can be done.

Overall, I find this study well executed, however, I think much can be done to clarify the results and their significance. I will also recommend further sequence analysis.

 We thank the reviewer for this overall positive statement. We agree that additional sequencing will add relevant information to the manuscript and we have included further full genome sequences in our analysis.

Line 52: Please check the sentence starting “Ticks paly the…” and rephrase.

 We agree that a word was missing in this sentence. The missing word “TBEV” was added.

Lines 65-67: How many cases are reported in the other states? I would suggest adding a map illustrating the variation in incidence/annual case numbers. I went on to check the database, but I found it quite tricky to find the actual data, but I do think this piece of information is essential for understanding the significance of these results.

We agree that the RKI database is not very intuitive to check the relevant numbers. Thus we provide the case numbers for all German federal states for the years 2001- 2018 in an additional FigureS1 and a TableS1

Methods/TBEV-isolates: It is a bit difficult to follow the isolation process, and what was exactly done on which tick pool, so please check the parts 2.2 and 2.3 concerning this. There are five pools positive for TBEV-RNA. For two isolates, you generated the whole genome. Why only two and why these two? Are these the two isolated elsewhere and what was the protocol used there? The three isolates described in this study were successful and from an isolate, the whole genome can and should be obtained (unless you can specifically state why not). Please also add a short description of the new sequences, how similar are they to each other and to other TBEV-Eur strains.

We added some text to the methods 2.2 section to clarify the isolation process for the different isolates. We agree that a full genome sequence of one of the other isolates will be very useful for an in depth interpretation. The two full genome sequences in the original manuscript were indeed the isolates cultivated by our partner in Munich. Both isolates were obtained from ticks from the focus Barsinghausen/Mooshütte. Both sequences were identical. Therefore, we choose to sequence one of our Isolates from the focus Rauher Busch and to compare the sequence to Barsinghausen/Mooshütte as well as to other EU-Isolates. We generated a new phylogenetic tree (Figure 1) including the new sequence and we added a table (Table 1) to compare the sequences Rauher Busch and Barsinghausen/Mooshütte.

Figure 1. Please show clearly the newly obtained sequences and other sequences from Germany as well.

The new sequences are marked in red and all German sequences are shown in bold letters.

Figure 2. This is a highly selected set of sequences, which may have an impact on the result. Please recalculate this phylogenetic tree with a larger collection of available E-gene sequences.

 We initially choose to show a reduced phylogenetic tree because we intended to show a more in detail resolution of phylogenetic relationships between E gene sequences that we show with the full genomes. In Figure 1 we show a the gross relationships between EU- isolates for TBEV and Figure 2 was meant as a zoom into German TBEV isolates to obtain a higher resolution. In the revised version of the manuscript, we now added an E-gene based phylogenetic tree including a large collection of available E Gene sequences.

Reviewer 2 Report

The manuscript "First isolation and phylogenetic analyses of tick-borne encephalitis virus in Lower Saxony, Germany" reports the finding of a new location for TBEV based on a field survey of ticks in the German federal state of Lower Saxony. The paper is well written and presented, with little major criticism and only a few minor corrections needed. 

Major point: The manuscript needs a map showing a number of features that are raised in the text. This includes the location of the survey sites, German federal states, neighbouring countries (especially Poland and Czech Republic) and the route of the motorways (E30 & E40). Without this it is difficult to visualise the conclusions drawn from the phylogeny.

Minor points:

Line 52: the sentence seems to be missing a word or phrase and does not make sense.

Line 118: Genbank accession numbers must be included in the final manuscript.

Line 168: the sentence "despite of two different sampling" needs revising.

References

9: include page numbers or equivalent.

15: its is not clear what this is? give more details and ensure it is accessible.

17: Provide full reference details

33: Duplication of e number?

34: Provide full reference details 

Author Response

The manuscript "First isolation and phylogenetic analyses of tick-borne encephalitis virus in Lower Saxony, Germany" reports the finding of a new location for TBEV based on a field survey of ticks in the German federal state of Lower Saxony. The paper is well written and presented, with little major criticism and only a few minor corrections needed. 

We thank the reviewer for this overall positive comment. We have taken the requested measures and adapted the manuscript according to the suggestions.

Major point: The manuscript needs a map showing a number of features that are raised in the text. This includes the location of the survey sites, German federal states, neighbouring countries (especially Poland and Czech Republic) and the route of the motorways (E30 & E40). Without this it is difficult to visualise the conclusions drawn from the phylogeny.

 We agree with  this reviewer that the relationships pointed out in the text are a bit difficult to follow up via Google maps or other maps. We therefor created Figure S2 which highlights the TBEV foci we obtained our isolates from and in addition the sampling spot for the other German isolates of which sequence information was used from using red circles. The motorway connection between Poland and Germany mentioned in the text are highlighted by dotted lines-

Minor points:

Line 52: the sentence seems to be missing a word or phrase and does not make sense.

We apologize for this mistake. Indeed a word was missing from this sentence which is now added.

Line 118: Genbank accession numbers must be included in the final manuscript.

We upload all new sequences to the NCBI database and will provide the respective accession numbers in the final manuscript proof.

Line 168: the sentence "despite of two different sampling" needs revising.

The respective sentence has been revised to clarify the point raised in this statement.

References

9: include page numbers or equivalent.

The page number was added.

15: its is not clear what this is? give more details and ensure it is accessible.

This is the database of the The Robert Koch Institute (RKI) for cases of notifiable diseases, and confirmation of pathogens, reported under the German ‘Act on the Prevention and Control of Infectious Diseases in Man’ (Infektionsschutzgesetz, IfSG). To facilitate the access to this database for our reader we added the URL for this website.

17: Provide full reference details

The reference was not fully published by the time we submitted this manuscript. Full reference details are now added.

33: Duplication of e number?

The duplication was removed

34: Provide full reference details 

Reference details including URL were added.

Round  2

Reviewer 1 Report

The authors have responded well to my comments and I recommend to accept the manuscript. Table 1 could be given as a supplementary figure. It will be interesting to see what happens in the future and will the human case numbers increase at some point. 

Author Response

We thank the reviewer for the positive response. We shifted table 1 to the supplement (now table S2).